# Correlation of Myocardial Native T1 and Left Ventricular Reverse Remodeling after Valvular Surgery

**DOI:** 10.3390/jcm12072649

**Published:** 2023-04-02

**Authors:** Maria von Stumm, Johannes Petersen, Martin Sinn, Theresa Holst, Tatiana M. Sequeira-Gross, Lisa Müller, Jonas Pausch, Peter Bannas, Gerhard Adam, Hermann Reichenspurner, Evaldas Girdauskas

**Affiliations:** 1Department for Congenital and Pediatric Heart Surgery, German Heart Center Munich, 80636 Munich, Germany; 2Department of Cardiovascular Surgery, University Heart & Vascular Center, University Medical Center Hamburg-Eppendorf, 20251 Hamburg, Germany; 3Department of Diagnostic and Interventional Radiology and Nuclear Medicine, University Medical Center Hamburg-Eppendorf, 20251 Hamburg, Germany; 4Department of Cardiac Surgery, University Clinic Augsburg, 86156 Augsburg, Germany

**Keywords:** left ventricular remodeling, myocardial native T1, T1 mapping, valvular cardiomyopathy, cardiac magnetic resonance imaging

## Abstract

Myocardial native T1 is a known cardiovascular magnetic resonance (CMR) imaging biomarker to quantify diffuse myocardial fibrosis in valvular cardiomyopathy. We hypothesized that diffuse myocardial fibrosis assessed by preoperative T1 mapping might correlate with LV reverse remodeling after valvular surgery. A prospective monocentric cohort study was conducted including 79 consecutive patients with valvular cardiomyopathy referred for surgical treatment of severe aortic or severe functional mitral regurgitation. Native T1 values were assessed by CMR before surgery. LV geometry parameters (i.e., LVEDV, LVESV) were obtained by 2D transthoracic echocardiography before and six months after surgery. Postoperative change of LV geometry parameters was calculated as delta (∆) variable (i.e., six months value minus baseline value). Mean native T1 was 1047 ± 39 ms, mean ∆LVEDV was −33 ± 42 mL, and mean ∆LVESV was −15 ± 27 mL. Native T1 values correlated with ∆LVEDV (Pearson r = 0.29; *p* = 0.009) and ∆LVESV (Pearson r = 0.29; *p* = 0.015). Native T1 values < 1073 ms were identified as independent predictor of postoperative reduction of LVEDV (HR 3.0; 95%-CI: 1.1–8.0; *p* = 0.03) and LVESV (HR 2.9; 95%-CI: 1.1–7.4; *p* = 0.03). Diffuse myocardial fibrosis assessed by myocardial native T1 correlates with LV reverse remodeling at six months after valvular surgery. T1 mapping may be a valuable tool to predict LV reverse remodeling in valvular heart disease.

## 1. Introduction

Heart valve regurgitation is a common disease in the Western world and its prevalence increases with an advancing age [1]. Even severe aortic or mitral regurgitation may remain asymptomatic for many years, although volume overload of the left ventricle (LV) leads to progressive LV remodeling. According to recent ESC/EACTS guidelines for the management of valvular heart disease, the indication for surgery of aortic regurgitation (AR) exists in symptomatic patients and in those with a progressive LV disease (i.e., LVEF < 50% or left ventricular end-systolic diameter (LVESD) > 50 mm) [2]. In case of functional mitral regurgitation (FMR), the indications for surgery are even more restrictive and are reserved for patients who remain severely symptomatic despite guideline-directed medical therapy and cardiac resynchronization therapy (CRT) [2].

Currently, the extent to which LV function and dimensions recover after successful valve repair is unknown. In theory, the myocardial damage should recede following correction of heart valve regurgitation. Practically, not all patients achieve postoperative LV reverse remodeling, depending on the degree of myocardial fibrosis [3,4]. In addition, persistence of postoperative LV dysfunction predicts a poor long-term prognosis [5].

The question of major clinical relevance is whether a more expeditious valvular surgery in case of severe asymptomatic aortic or mitral valve regurgitation may prevent the progression of LV remodeling at an early stage. Therefore, more sensitive biomarkers would be required to better and earlier quantify the ongoing myocardial damage that may precede a significant LV enlargement.

Current cardiac magnetic resonance (CMR) quantitative mapping techniques show a strong correlation between native T1 and histologic collagen volume fraction, which determines the degree of myocardial fibrosis [6]. A previous consensus statement underlined the value of native T1 for visualization and quantification of diffuse myocardial fibrosis [7]. However, myocardial tissue analysis by CMR currently has no impact in the decision-making process regarding the timing of valvular surgery.

We hypothesized that diffuse myocardial fibrosis defined by T1 mapping in the preoperative CMR may correlate with LV reverse remodeling defined by a reduction of LVEDV and LVESV at six months after valvular surgery.

## 2. Materials and Methods

### 2.1. Study Design

The institutional ethics committee approved this prospective study, and all subjects gave written informed consent (Ethical Committee of Medical Council, Hamburg, Germany; PV5382).

Consecutive patients with severe AR or severe FMR referred to our institution for elective valvular surgery between July 2017 and August 2019 were prospectively enrolled in this study. Study exclusion criteria were as follows: predominant aortic or mitral valve stenosis, valvular redo procedures, common contraindications for MRI such as severe obesity (250 kg CMR table weight limit), and presence of a metallic foreign bodies.

The severity of valve regurgitation was defined by established diagnostic criteria, including transthoracic echocardiography and clinical aspects [2].

Study protocol imaging consisted of preoperative CMR with myocardial native T1 mapping and transthoracic 2D-echocardiography with measurement of LV size parameters: LV end-diastolic diameter (LVEDD), LV end-systolic diameter (LVESD), LV end-diastolic volume (LVEDV), LV end-systolic volume (LVESV), and LV ejection fraction (LVEF). Follow-up echocardiographic re-evaluation was performed at 6-months after surgery.

In addition, laboratory values of serum N-terminal prohormone of brain natriuretic peptide (NT-proBNP; pg/mL) and serum creatinine (mg/dL) were preoperatively assessed.

### 2.2. Assessment of Cardiac Magnetic Resonance Imaging

Preoperative CMR was routinely performed within one week prior to surgery. All cardiac examinations were conducted by a 1.5-T MR scanner (Achieva; Philips Medical Systems, Best, The Netherlands). CMR was performed using a standardized protocol including an electrocardiogram-gated, standard steady-state, free-precession cine MR sequence, acquired during breath holds in standard long-axis views (4-, 3-, and 2-chamber view) and short-axis slices covering the entire left ventricle [8]. Native T1 mapping was acquired using a 5 s (3 s) 3 s modified Look-Locker inversion-recovery (MOLLI) sequence on three short-axis sections (apical, middle, and basal). Each CMR set was independently analyzed by two investigators. Mapping values were measured by drawing a single region of interest (ROI) in the septum on mid-cavity short-axis maps.

Native T1 measurements showed good inter-observer reliability (intraclass correlation coefficient: absolute value 0.895; 95% confidence interval 0.825–0.937).

### 2.3. Assessment of Echocardiographic Left Ventricular Geometry Parameters

Baseline and follow-up 2D transthoracic echocardiography images were routinely recorded and analyzed in our echocardiographic core laboratory by two independent investigators. Baseline echocardiographic examination was conducted within one week prior to surgery. Follow-up echocardiographic examination was conducted six months after surgery.

Using a standardized protocol, LV geometry parameters, including LVEF, LVEDD, LVESD, LVEDV, and LVESV, were measured in the 4- and 2-chamber view during end-systole or end-diastole according to the Recommendations for Cardiac Chamber Quantification by Echocardiography in Adults from the American Society of Echocardiography and the European Association of Cardiovascular Imaging [9]. The method for 2D echocardiographic volume calculations was the biplane method of disks summation (modified Simpson’s rule) [9]. Missing follow-up values of LVEDV and LVESV were calculated using Teichholz formula based on LVEDD and LVESD [10].

To generate high accuracy and reproducibility of biplane two-dimensional echocardiographic measurements, the same investigators followed the same patients at baseline and follow-up as recommended by Otterstad et al. [11] Our study followed a very similar echocardiographic measurement protocol as compared to Baron et al. [12]; therefore, we expected comparable high test-retest reliability values for the echocardiographic LV measurements.

### 2.4. Surgical Technique

Depending on the underlying disease (i.e., AR or FMR) surgical strategies differed. Surgical approach for AR cases were partial sternotomy or median sternotomy depending on the need of concomitant cardiac procedures and patients’ comorbidities. Treatment strategies for AR patients ranged from aortic valve repair to biological or mechanical aortic valve replacement.

Surgical approach to FMR was either median sternotomy or right antero-lateral mini-thoracotomy. FMR was treated using a standard rigid complete annuloplasty ring, which was downsized by one size, according to the length of the anterior mitral leaflet. In case of severe leaflet tenting, subannular repair with repositioning of both papillary muscles was performed [13]. There were no major variations in the surgical or perioperative management during the study period.

### 2.5. Statistical Analysis

Normally distributed continuous variables are presented as mean ± standard deviation and categorical variables are expressed as percentages throughout the manuscript. Comparison of normally distributed continuous variables was performed by unpaired *t*-test. Non-normally distributed, unpaired data was compared using the Mann–Whitney-U test. Fisher’s exact test was used for univariable comparisons of categorical variables. A Pearson correlation coefficient was used for correlation analyses.

The difference of LVEF, LVEDD, LVESD, LVEDV, and LVESV at six months’ follow-up and at baseline were calculated as delta (∆) variable (i.e., follow-up value minus baseline value). In line with this, negative ∆ values of LVEDD, LVESD, LVEDV, and LVESV indicated postoperative reduction of LV geometry parameters and presence of LV reverse remodeling. While positive ∆ values of LVEDD, LVESD, LVEDV, and LVESV indicated increase of postoperative LV geometry parameters and absence of LV reverse remodeling. For ∆LVEF applied the opposite, negative ∆LVEF values indicated postoperative decrease of LVEF and absence of LV reverse remodeling, while positive ∆LVEF indicated presence of LV reverse remodeling.

We calculated receiver operating characteristic (ROC) curves to determine optimal cut-off points of myocardial native T1 values to predict postoperative reduction in LV geometry parameters. The optimal cut-off value was defined by Youden index. Predictors of reduction in LV geometry parameters were subsequently assessed by multivariable Cox regression analysis. All *p*-values < 0.05 were considered statistically significant. All statistical analyses were performed using the SPSS version 26.0 statistical package (IBM Corp., Markham, ON, Canada).

### 2.6. Study Endpoint

Primary study endpoint was the correlation between baseline T1 values and LV reverse remodeling defined by reduction of LVEDV and LVESV at six months postoperatively after valvular surgery.

## 3. Results

### 3.1. Demographics and Baseline Characteristics

A total of 79 consecutive patients (mean age 56 ± 14 years) with severe AR (*n* = 46) or severe FMR (*n* = 33) were included in our study (see Table 1).

Overall, there were 58/79 male (73%) and 36/79 female (27%) participants. The original aortic valve pathology included unicuspid aortic valve disease in five cases (11%) and bicuspid aortic valve disease in 16 cases (35%). All other AR patients presented with tricuspid aortic valves (*n* = 25; 54%). Concomitant root enlargement was found in 11 cases (24%).

Depending on the underlying disease, operative procedures consisted of aortic valve repair (*n* = 27), or aortic valve replacement (*n* = 19), or mitral valve repair using ring annuloplasty combined with subannular repair (*n* = 20) or isolated ring annuloplasty (*n* = 13). In the FMR cohort, concomitant surgeries included coronary artery bypass surgery (*n* = 13) and tricuspid valve repair (*n* = 5). None of the thirteen patients who underwent simultaneously CABG showed a regional myocardial fibrosis corresponding to the diseased coronary artery territory in CMR.

During the follow-up period, four re-operations were required, including endocarditis following aortic valve replacement at six months follow-up (*n* = 1), recurrence of AR after aortic valve repair due to suture dehiscence at eight months follow-up (*n* = 1), and recurrent severe FMR following isolated ring annuloplasty at one week follow-up and one month follow-up (*n* = 2). Echocardiographic grading of recurrent valve regurgitation at six months follow-up are depicted in Table 1. There were no deaths registered during the follow-up period.

### 3.2. Imaging Data and Correlation between T1 Values and Echocardiographic LV Measurements

The comparison of echocardiographic LV measurements at 6 months postoperatively vs. baseline revealed mean ∆LVEF of 1 ± 8%, mean ∆LVEDD −6 ±7 mm, mean ∆LVESD −5 ±7 mm, mean ∆LVEDV of −33 ± 42 mL, and mean ∆LVESV of −15 ± 27 mL. A reduction of diastolic and systolic LV volumes was found in 58 patients (73%) and in 56 patients (71%), respectively.

The myocardial native T1 values of the study cohort were normally distributed with a mean baseline native T1 of 1047 ± 39 ms (see Figure 1).

ROC curve analysis revealed an optimal cut-off point of myocardial native T1 of 1073 ms to predict reduction of LVEDV (sensitivity 85%, 95%-CI: 74–95; specificity 57%, 95%-CI: 29–85; area under the curve (AUC) 0.64, 95%-CI: 0.51–0.78) and LVESV (sensitivity 80%, 95%-CI: 67–83; specificity 60%, 95%-CI: 32–88; AUC 0.64, 95%-CI: 0.51–0.78). Calculated in a Cox regression model using covariables age, gender, and level of creatinine, native T1 < 1073 ms was identified as an independent predictor for reduction of LVEDV (HR 3.0; 95%-CI: 1.1–8.0; *p* = 0.03) and LVESV (HR 2.9; 95%-CI: 1.1–7.4; *p* = 0.03) (Table 2).

### 3.3. Post-Hoc Subgroup Analysis by Valve Pathology

Comparison of the baseline characteristics between AR patients (*n* = 46) vs. FMR patients (*n* = 33) revealed significantly younger age (AR: 51.4 ± 14.2 years vs. FMR: 61.6 ± 12.1 years; *p* = 0.001), lower values of preoperative NT-proBNP (AR: 801 ± 1633 pg/mL vs. FMR: 3715 ± 3483 pg/mL; *p* < 0.001), and serum creatinine (AR: 1.0 ± 0.4 vs. FMR: 1.3 ± 0.6 mg/dL; *p* = 0.02) in the AR subgroup (Table 1). In line with this, preoperative LVEF was significantly higher in AR patients vs. FMR patients (AR: 57 ± 8% vs. FMR: 37 ± 11%; *p* < 0.001). Similar, preoperative CMR revealed significantly lower native T1 values in AR vs. FMR patients (AR: 1032 ± 32 ms vs. FMR: 1066 ± 39 ms; *p* < 0.001).

Systolic echocardiographic LV geometry indexes including LVESD (AR: 41 ± 9 mm vs. FMR 50 ± 9 mm; *p* < 0.001) and LVESV (AR: 74 ± 31 mL vs. FMR: 110 ± 40 mL; *p* < 0.001) were significant lower in AR patients vs. FMR patients at baseline. After six months, reduction of LVEDV and LVESV was found in 37 (80%) and 35 (75%) AR patients vs. 21 (64%) and 21 (64%) FMR patients, respectively. Therefore, contribution of the primary study endpoint was comparable between both study cohorts (reduction of LVEDV: AR: 80% vs. FMR: 64%, *p* = 0.43; reduction of LVESV: AR: 75% vs. FMR: 64%, *p* = 0.12). Although, the mean ∆LVEDV improved more extensively in AR vs. FMR patients (as indicated by postoperative LVEDV reduction of −44 ± 45 mL in AR patients vs. −19 ± 34 mL in FMR patients, *p* = 0.01). Of note, there were no statistically significant difference between AR and FMR patients regarding the postoperative reduction of LVESV (i.e., −17 ± 24 mL in the AR cohort vs. −13 ± 30 mL in the FMR cohort; *p* = 0.5).

## 4. Discussion

In our study, we aimed to evaluate the relationship between preoperative T1 values as defined by CMR, and LV reverse remodeling in valvular cardiomyopathy. We were able to demonstrate that native T1 values may quantify a diffuse myocardial fibrosis and correlate with the change in LVEDV and LVESV between baseline and 6-months follow-up after valvular surgery. Particularly, shorter preoperative native T1 times (i.e., <1073 ms) were associated with an increased prevalence of LV reverse remodeling following valvular surgery. Based on these findings, we conclude that preoperative native T1 values might be a valuable tool to predict reverse LV remodeling after surgery for aortic or mitral regurgitation. Furthermore, shorter native T1 times may indicate a reversible myocardial dysfunction, which improves after valve surgery, while higher T1 values could imply a rather fixed defect.

Post-hoc subgroup analysis revealed significantly longer T1 times in FMR patients indicating a more severe LV disease in FMR patients in comparison to AR patients. Accordingly, FMR patients showed lower degree of postoperative LV reverse remodeling. These findings confirm our clinical observation, that FMR patients present mostly with a more advanced LV disease as compared to AR patients.

Native T1 mapping has previously been proposed to be a valuable diagnostic marker for several cardiac pathologies [14,15,16,17,18]. However, native T1 measurements vary between CMR sites due to several technical factors and values are not standardized. In line with this, the identified cut-off point of native T1 of 1073 ms of our study can be used as rough orientation. In line with this, it should not be generalized as reference value for other CMR sites.

Currently, there is a lack of robust evidence from adequately powered clinical trials on the role of fibrosis variables in informing clinical decisions for surgical or catheter-based interventions. If confirmed by future studies, our findings may have important clinical implications. By means of native T1 mapping, risk stratification and treatment selection in patients with valvular cardiomyopathy may be further improved. Especially, timing of surgical intervention may be optimized before severe myocardial damage occurs. In line with this, patients with valvular heart and an irreversible myocardial defect may benefit from less invasive treatment strategies (i.e., transcatheter procedures).

Unfortunately, the current treatment recommendations for AR or FMR of the ESC/EACTS guidelines of management valvular heart disease do only refer to LVEF < 50% and LVESD > 50 mm (LVESDi > 25 mm/m^2^) as a trigger for valvular intervention [2]. In the guidelines, neither the impact of diffuse myocardial fibrosis diagnosed by T1 mapping during CMR nor the effects of LV reverse remodeling on the outcomes following valvular surgery are further discussed [2]. Repeated measurements of native T1 values in asymptomatic patients with left-sided heart valve insufficiency may support individual decision making in appropriate timing of valvular intervention. Such strategy could prevent irreversible myocardial damage.

Our study possesses all the limitations of a single-center experience with a relatively low-numbered study cohort.


(1)Our study cohort consisted of patients with different valve pathologies including tricuspid (TAV) and bicuspid (BAV) aortic valve disease and functional mitral valve disease. This could be a confounder to our analysis. However, the development of diffuse myocardial fibrosis and valvular cardiomyopathy has the same pathophysiological background in both TAV and BAV patients. Chronic volume overload results in eccentric LV hypertrophy and LV remodeling, which is independent of valve morphology. Due to the common pathophysiological pathway in TAV and BAV, we analyzed both valve pathologies as a uniform entity, the AR subgroup. Furthermore, we aimed to demonstrate that the correlation between baseline T1 and LV reverse remodeling after valvular surgery is independent of the type of valvular disease (i.e., AR vs. FMR). To evaluate this assumption, we included valve type (i.e., AR vs. FMR) into our Cox regression model. Cox regression model confirmed the correlation between native dichotomized T1 and LV reverse remodeling to be independent of the type of valvular disease (i.e., AR vs. FMR). In other words, the correlation between native T1 mapping and post-surgical LV reverse remodeling was comparable between AR and FMR patients.(2)Our study cohort included thirteen study patients, who underwent simultaneous CABG surgery. Even though the presence of CAD was the main pathophysiological mechanism leading to cardiomyopathy, none of them had a regional myocardial fibrosis corresponding to the diseased coronary artery territory. Therefore, this patient cohort with CAD represents a very selective patient subgroup with predominantly diffuse myocardial fibrosis.(3)Four patients underwent redo surgery during the study period, which might have influenced the LV reverse remodeling course as defined by reduction of LVEDV and LVESV. However, the redo surgeries were performed at the first month or after the 6 months follow-up echocardiographic examination. Therefore, in all four cases, there was presumably enough time for the LV to remodel.(4)The ROC analyses of our study, which we performed to determine the cut-off point of the prediction variable native T1, revealed only acceptable AUC values for reduction of LVEDV and LVESV. In addition, the confidence intervals of AOCs showed a wide range of values due to the low number of study patients.(5)Analysis and interpretation of echocardiographic measurements of LV remodeling parameters relied on the test-retest variability published by Baron et al. [12].


In summary, our current findings need to be confirmed by a consecutive prospective large multicenter study.

## 5. Conclusions

Diffuse myocardial fibrosis assessed by myocardial native T1 correlates with LV reverse remodeling at six months after valvular surgery. T1 mapping may be a valuable tool to predict LV reverse remodeling in valvular heart disease; however, this should be validated in a larger prospective multicenter study.

## Figures and Tables

**Figure 1 jcm-12-02649-f001:**
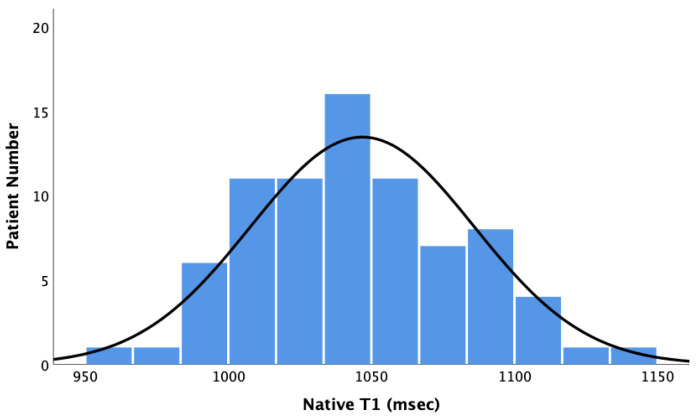
Bar Chart showing the distribution of native T1 values. Baseline native T1 values did not correlate with ∆LVEF (Pearson r = −0.002, *p* = 0.98), ∆LVEDD (Pearson r = 0.02, *p* = 0.84), and ∆LVESD (Pearson r = −0.05, *p* = 0.71). However, baseline native T1 values correlated with ∆LVEDV (Pearson r = 0.291, *p* = 0.009) and ∆LVESV (Pearson r = 0.292, *p* = 0.015) (see Figure 2 and Figure 3).

**Figure 2 jcm-12-02649-f002:**
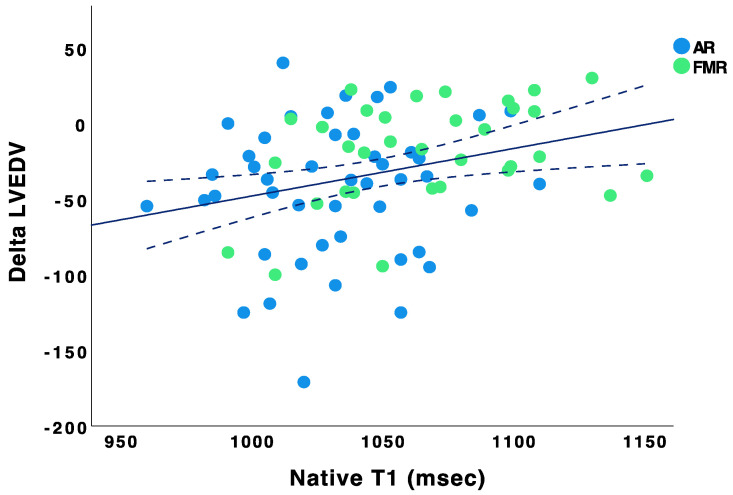
Correlation of native T1 with ∆LVEDV. Scatter plot showing correlation of myocardial native T1 with ∆LVEDV (Pearson r = 0.291, *p* = 0.009). ∆LVEDV was calculated by six months value of LVEDV (mL) *minus* baseline value of LVEDV (mL). AR patients are blue and FMR patients are green marked.

**Figure 3 jcm-12-02649-f003:**
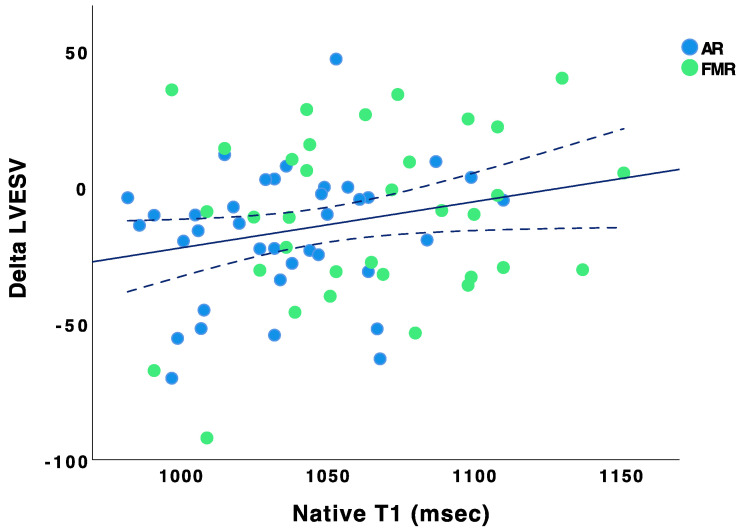
Correlation of native T1 with ∆LVEDV. Scatter plot showing correlation of myocardial native T1 with ∆LVESV (Pearson r = 0.292, *p* = 0.015). ∆LVESV was calculated by six months value of LVESV (mL) *minus* baseline value of LVESV (mL). AR patients are blue and FMR patients are green marked.

**Table 1 jcm-12-02649-t001:** Demographics and baseline characteristics.

Patient Characteristics	All (*n* = 79)	AR Subgroup (*n* = 46)	FMR Subgroup (*n* = 33)	*p* Value AR vs. FMR
Male (%)	58 (73)	36 (78)	22 (67)	0.25
Age (years)	55.7 ± 14.2	51.4 ± 14.2	61.6 ± 12.1	0.001
BSA (m^2^)	2.0 ± 0.2	2.1 ± 0.2	1.9 ± 0.2	0.007
Comorbidities				
Diabetes (%)	6 (8)	2 (4)	4 (13)	0.17
Hypertension (%)	39 (51)	26 (57)	13 (42)	0.20
CAD (%)	23 (30)	2 (4)	21 (68)	<0.001
STS Score *	1.0 ± 0.6	1.0 ± 0.6	1.0 ± 0.6	0.91
EuroSCORE II *	1.2 ± 0.8	1.1 ± 0.8	1.5 ± 0.8	0.05
Creatinine (mg/dL)	1.2 ± 0.5	1.0 ± 0.4	1.3 ± 0.6	0.02
NT-proBNP (pg/mL) *	2015 ± 2927	801 ± 1633	3715 ± 3483	<0.001
Baseline imaging values				
Native T1 (ms)	1047 ± 39	1032 ± 32	1066 ± 39	<0.001
LVEF (%) *	49 ± 13	57 ± 8	37 ± 11	<0.001
LVEDD (mm)	61 ± 9	60 ± 9	62 ± 8	0.27
LVESD (mm)	46 ± 10	41 ± 9	50 ± 9	<0.001
LVEDV (mL)	170 ± 54	164 ± 55	177 ± 52	0.30
LVESV (mL) *	90 ± 39	74 ± 31	110 ± 40	<0.001
Change of LV geometry indexes after 6 months				
∆LVEF (%)	1 ± 8	−1 ± 7	2 ± 10	0.22
∆LVEDD (mm) *	−6 ± 7	−7 ± 7	−3 ± 5	0.011
∆LVESD (mm)	−5 ± 7	−5 ± 7	−5 ± 8	0.99
∆LVEDV (mL) *	−33 ± 42	−44 ± 45	−19 ± 34	0.010
∆LVESV (mL)	−15 ± 27	−17 ± 24	−13 ± 30	0.54
LV reverse remodeling after 6 months				
∆LVEDV < 0 (%)	58 (73)	37 (80)	21 (64)	0.09
∆LVESV < 0 (%)	56 (71)	32 (70)	24 (73)	0.37
Valve regurgitation after 6 months				
None (%)	33 (42)	19 (41)	14 (42)	NA
Mild (%)	43 (54)	26 (57)	17 (52)	NA
Moderate (%)	1 (1)	1 (2)	0	NA
Severe (%)	2 (3)	0	2 (6)	NA

Delta (∆) variable was calculated by six months value *minus* baseline value of LV geometry indexes. With asterix * marked parameters were compared by Mann–Whitney U test.

**Table 2 jcm-12-02649-t002:** Predictors of LV reverse remodeling.

	ß	*p* Value	Hazard Ratio	Confidence Interval
Predictors of LV reverse remodeling measured by LVEDV (∆LVEDV < 0)				
Age	0.03	0.20	1.0	0.99–1.07
Gender	−0.88	0.06	0.41	0.17–1.04
Creatinine Level (mg/dL)	−0.21	0.53	0.81	0.43–1.55
Native T1 < 1073 ms	1.10	0.03	3.00	1.10–8.00
Valve type (AR vs. MR)	0.37	0.48	1.45	0.53–3.99
Predictors of LV reverse remodeling measured by LVEDV (∆LVEDV < 0) (unadjusted model)				
Native T1 < 1073 ms	0.99	0.02	2.69	1.13–6.39
Predictors of LV reverse remodeling measured by LVESV (∆LVESV < 0)				
Age	0.01	0.61	1.01	0.97–1.05
Gender	−0.91	0.05	0.40	0.16–1.01
Creatinine Level (mg/dL)	−0.61	0.17	0.54	0.22–1.29
Native T1 < 1073 ms	1.05	0.03	2.90	1.10–7.42
Valve type (AR vs. MR)	0.50	0.36	1.64	0.57–4.72
Predictors of LV reverse remodeling measured by LVESV (∆LVESV < 0) (unadjusted model)				
Native T1 < 1073 ms	0.89	0.03	2.43	1.05–5.64

Independent predictors of the LV reverse remodeling were assessed by multivariable analysis using the Cox proportional hazards regression model. Delta (∆) variable was calculated by six months value *minus* baseline value of LV geometry indexes.

## Data Availability

The data underlying this article will be shared on reasonable request to the corresponding author.

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
