# Peer review of "Correlation of Myocardial Native T1 and Left Ventricular Reverse Remodeling after Valvular Surgery"

_jcm, 2023, doi:10.3390/jcm12072649_

Round 1

Reviewer 1 Report (New Reviewer)

Dear authors, I have studied with great interest the manuscript "Correlation of native myocardial T1 and left ventricular Reverse remodeling after valve surgery"

The method of magnetic resonance imaging (MRI) is considered the gold standard in assessing the functional state of the heart, as well as visualization of the myocardium. MTI, along with the traditional imaging method - EchoCG, also plays a leading role in the diagnosis of valvular heart disease and assessing the effectiveness of surgical intervention for this pathology. The results of several small studies suggest that, compared with echocardiography, MRI provides greater accuracy in terms of quantifying the volume and mass of LV lesions and predicting outcomes after surgical treatment of valvular heart disease, including depending on the type of intervention performed.

The authors convincingly showed that MRI using the technique of T1-mapping of the myocardium, quantitatively determining diffuse myocardial fibrosis, makes it possible to characterize in detail the processes of reverse cardiac remodeling. The results of the study have the potential to develop approaches to predict outcomes after heart valve surgery using MRI, as it has been shown that diffuse myocardial fibrosis, assessed using preoperative T1 mapping, can correlate with reverse LV remodeling after valve surgery. This is confirmed in studies by other scientists, in particular, it is known that fibrosis of the middle segments of the left ventricle, established by MRI, can be considered as an early marker of irreversible decompensation of heart failure, as well as an independent predictor of all-cause mortality and cardiovascular death.

I express my gratitude to the authors for their work and my great pleasure in reading their results.

Author Response

Thank you very much for your review.

There were no comments or changes required. 

Reviewer 2 Report (New Reviewer)

I have reviewed the manuscript "Correlation of Myocardial Native T1 and Left Ventricular Reverse Remodeling after Valvular Surgery" by von Stamm et al. In this study the authors want to study the correlation between CMR T1 values (reflecting fibrosis of the LV) to the potential of LV reverse remodeling after aortic- and mitral valve surgery due to regurgitation. The idea is that T1 could be a new biomarker to better predict optimal timing of surgery for AR and MR. The hypothesis is very appealing and has a direct clinical implication. Unfortunately, there are several concerns that should be addressed and corrected:

1) The Aortic Regurgitation patients represent a mixed cohort, which is composed of 35% BAV and 11% unicuspid valves and rest tricuspid aortic valves. We know from several other publications that BAV patients are much younger than corresponding TAV patients in need of surgery. BAV is also associated to aortopathy as well as left ventricular hypertrophy. Most certainly, unicuspid patients have similar characteristics. Therefore, in order to have a uniform AR cohort, BAV and unicuspid patients should be excluded or the different subgroups presented separately. BAV and unicuspid patients can be presented as one cohort, separated from the TAV counterparts. 

Also, coronary artery diseased patients should be excluded, since CAD predisposes to fibrosis by itself and will be a confounder to the analysis.

2) The mitral regurgitation is stated to be a non-ischemic functional MR. That is not true since 68% of the patients have CAD and CABG in performed in 39% of MR patients. These patients should be excluded from analysis.

3) In the AR patients, the majority of the patients underwent plasty of the the valves. Since there exists an increased risk of recurrent AR during follow-up, the grading of AR at 6 months should be registered in the table.

4) The same as for the AR, MR patients underwent plasty and the grade of MR at follow-up should be registered in table. 

Author Response

Reviewer 2

I have reviewed the manuscript "Correlation of Myocardial Native T1 and Left Ventricular Reverse Remodeling after Valvular Surgery" by von Stamm et al. In this study the authors want to study the correlation between CMR T1 values (reflecting fibrosis of the LV) to the potential of LV reverse remodeling after aortic- and mitral valve surgery due to regurgitation. The idea is that T1 could be a new biomarker to better predict optimal timing of surgery for AR and MR. The hypothesis is very appealing and has a direct clinical implication. Unfortunately, there are several concerns that should be addressed and corrected:

Comment 1: The Aortic Regurgitation patients represent a mixed cohort, which is composed of 35% BAV and 11% unicuspid valves and rest tricuspid aortic valves. We know from several other publications that BAV patients are much younger than corresponding TAV patients in need of surgery. BAV is also associated to aortopathy as well as left ventricular hypertrophy. Most certainly, unicuspid patients have similar characteristics. Therefore, in order to have a uniform AR cohort, BAV and unicuspid patients should be excluded or the different subgroups presented separately. BAV and unicuspid patients can be presented as one cohort, separated from the TAV counterparts. Also, coronary artery diseased patients should be excluded, since CAD predisposes to fibrosis by itself and will be a confounder to the analysis.

Answer 1: We share the reviewers concerns about pathophysiological differences in aortic valve disease between BAV and TAV patients, and the clinical implications of coronary artery disease on the presence of myocardial fibrosis.

1) Even though demographic and phenotypic differences exist between TAV and BAV patients, the development of diffuse myocardial fibrosis and valvular cardiomyopathy has the same pathophysiological background in both AR cohorts. This is a chronic volume overload resulting in eccentric LV hypertrophy and LV remodelling which is independent of valve morphology (i.e., BAV or TAV). Due to the common pathophysiological pathway in the AR cohort, we decided to analyse the AR cohort as a uniform entity leading to volume-overload induced valvular cardiomyopathy. We added this explanatory information to the Discussion section.   

(2) The presence of CAD was the main pathophysiological mechanism leading to cardiomyopathy in those thirteen study patients who underwent simultaneous CABG surgery, none of them had a regional myocardial fibrosis corresponding to the diseased coronary artery territory. Therefore, this patient cohort with CAD represents a very selective patients’ subgroup with predominantly diffuse myocardial fibrosis. We included this information into the Limitation section.

Furthermore, by means of native T1 mapping we focussed specifically on the diffuse myocardial fibrosis component (and not on the total fibrosis burden) and aimed to evaluate the correlation between baseline T1 and LV reverse remodeling after valvular surgery. We expected this correlation to be the function of the severity of diffuse myocardial fibrosis and independent of the type of valvular disorder or the underlying pathogenetic mechanism. To evaluate this assumption, we included valve type (i.e., AR vs. FMR) into our established Cox regression model. Cox regression model confirmed the correlation between native dichotomized T1 and LV reverse remodelling to be independent of the type of valvular disease (i.e., AR vs. FMR). In other words, the correlation between native T1 mapping and post-surgical LV reverse remodelling is comparable between AR and FMR patients.  We included this information into the Discussion section.

Comment 2: The mitral regurgitation is stated to be a non-ischemic functional MR. That is not true since 68% of the patients have CAD and CABG in performed in 39% of MR patients. These patients should be excluded from analysis.

Answer 2: Please see Answer 1). In addition, we reworded the term “non-ischemic functional MR” into “functional MR” only.

Comment 3: In the AR patients, the majority of the patients underwent plasty of the the valves. Since there exists an increased risk of recurrent AR during follow-up, the grading of AR at 6 months should be registered in the table.

Answer 3:

A single patient had a recurrent AR grade 2 at 6 months’ follow-up after aortic valve repair surgery and underwent redo surgery due to cusp suture dehiscence two months later. All remaining 26 patients had a stable echocardiographic AV repair result at 6 months’ follow-up: no residual AR in 10 patients and mild AR in the remaining 16 patients. We included the echocardiographic 6 months’ follow-up results in the revised manuscript at Table 2.

Comment 4: The same as for the AR, MR patients underwent plasty and the grade of MR at follow-up should be registered in table. 

Answer 4: In the MR cohort, two patients had severe recurrent MR after an isolated ring annuloplasty and underwent redo surgery subsequently. The first patient underwent redo surgery one week after isolated ring annuloplasty and the second patient after one month. The remaining MR patients had stable echocardiographic outcome: 14 patients had no residual MR and the remaining 17 patients had mild residual MR. We included the echocardiographic 6 months’ follow-up results in the revised manuscript at Table 2.

Reviewer 3 Report (New Reviewer)

Maria von Stumm and coworkers investigated the association of myocardial fibrosis assessed by MRI with LV reverse  remodeling after valvular surgery in 79 consecutive patients with valvular cardiomyopathy referred for surgical treatment of severe aortic or severe functional mitral regurgitation. They found that a low level of fibrosis was a significant independent predictor of LV recovery.

Specific comments

The definition of the primary endpoint is vague. It is stated that the primary endpoint was the correlation between baseline T1 values and LV reverse remodeling. However, there were 2  variables to characterize LV remodeling.

We need more information on the multivariable Cox-Models. How were the variables chosen? Obviously, the models need to be highly parsimonious because of the low number of patients included.

How did the authors arrive at the current sample size.

The handling of the four patients with re-operations needs to be addressed and discussed as a limitation.

I find the analysis according to valve pathology problematic. On the one hand, it is clear that the 2 pathologies differ substantially (primarily a  valvular versus myocardial pathology) and they have little in common except volume overload. On the other hand, figures 3 and 4 nicely show that the relation between fibrosis and remodeling was similar in the two entities. Instead of the tedious presentation of this matter in the Results, I suggest a simple Cox model with two independent variables –dichotomized T1 and valve type – plus an interaction term -dichotomized T1 x valve type.

Please, add the crude HRs and CI for dichotomizedT1 in Table 2.

The authors state, “Our study has important potential clinical implications.” I suggest rephrasing, “If confirmed by future studies our findings may have important clinical implications.” I find the whole paragraph starting with this sentence speculative and over-enthusiastic. Please, tone down. With regard to the guidelines, please keep in mind that there is a lack of robust evidence from adequately powered clinical trials on the role of fibrosis variables in informing clinical decisions for surgical or catheter-based interventions.

The low coefficients of variation are a limitation for the ultimate goal of informing clinical decisions. This needs to be addressed.  In this respect, the AUCs of the ROC analyses should also be mentioned and discussed.

Round 2

Reviewer 2 Report (New Reviewer)

Dear authors

I believe the manuscript has improved thanks to the revisions performed. I congratulate you to a well performed study which together with future studies with make a base for using CMR in the decision-making process. 

This manuscript is a resubmission of an earlier submission. The following is a list of the peer review reports and author responses from that submission.

Round 1

Reviewer 1 Report

Manuscript ID: jcm-2061551

Title: Correlation of myocardial native T1 and left ventricular reverse remodeling after valvular surgery

Comments to the authors

Manuscript provides very interesting data but it still needs a considerable revision to be acceptable for JCM. I have a few questions and would like to be clarified.

1. There are two sections of Discussion, and some of the contents are overlapped and difficult to understand. I think it should be summarized.

2. Since the mechanism of myocardial damage seems to be different between AR and MR, Figures 3 and 4 should be described by etiology of valvular disease.

3. R = 0.29 seems to have a fairly weak correlation. Not remarkably.

Author Response

Reviewer # 1 

Comment 1: There are two sections of Discussion, and some of the contents are overlapped and difficult to understand. I think it should be summarized.

Answer 1: We corrected this and removed the overlapping parts of the Discussion section.

Comment 2: Since the mechanism of myocardial damage seems to be different between AR and MR, Figures 3 and 4 should be described by etiology of valvular disease.

Answer 2: We assumed that the pathophysiological mechanism of valvular cardiomyopathy was comparable in both study cohorts. In both study subgroups volume-overload seems to be the main pathophysiological mechanism leading to cardiomyopathy.  

To demonstrate the distribution of AR and FMR in the Figure 3 and 4, we use now different colours.

Comment 3: R = 0.29 seems to have a fairly weak correlation. Not remarkably.

Answer 3: We agree with you. However, as mentioned in the limitation section, our study cohort was relatively low-numbered. In addition, our follow-up period was limited to six months, while left ventricular reverse remodeling may be an ongoing process, which takes longer time periods. Therefore, longer follow-up times can be awaited to strengthen this correlation.

Reviewer 2 Report

Here von Stumm et al. describe the correlation of preoperative T1 mapping in aortic regurgitation (AR) and functional non-ischemic mitral regurgitation (FMR) against cardiac remodeling after surgery. They found a correlation of the higher the T1 mapping (as a predictor of myocardial fibrosis) preoperative, the less improvement of LV geometry parameters after surgery.

Overall an interesting paper, as cardiovascular magnetic resonance tomography (CMR) is a good non-invasive approach for the optimal decision-making of the timepoint for a required surgery.

However, I have a some comments:

11.       Materials & Methods:

a.       Line 70-71: a short definition of what is meant by severe obesity would be nice. Is it a specific cut-off at e.g. BMI > 40 or does the CMR have a weight limit?

b.       Line 86 ff: When was the time point of the CMR, as it was mentioned for the echo but not for the CMR itself.

c.       Line 94: You mention one independent investigator, why not two as you did for echo?

d.       Line 125 ff: You mentioned using the unpaired t-test for normally distributed values, but did you test for the distribution? And were all data normally distributed? If not, what test was used then?

e.       Line 131: It is nice to have a clear description of what the delta values of geometry parameters mean, however, for the LVEF it is the opposite way, which would be nice to be clarified too (mean a negative delta in LVEF indicated a decreasing LVEF).

22.       Results:

a.       Line 149 ff (paragraph 3.1): As you mentioned two independent investigators for the echo data, an interpreter variability should be mentioned.

b.       Line 211 ff (paragraph 3.3): I am not convinced, that this section is a real subgroup analysis for this study. You mentioned the differences in the parameters between AR and FMR, but there is no distinction between these groups regarding the correlation from T1 to the delta measurements. I would like to see an analysis as you did in paragraph 3.2 for AR and FMR separately.

33.       Discussion:

a.       It is quite sad that through the reading process and approval from all authors of the submitted version, no one saw, that lines 233-269 are exactly the same as 273-309

b.       Furthermore, lines 270-272 are information from the template and do not relate to the paper.

c.       Line 284: You mentioned that T1 values > 1073ms as “rather fixed defects”. I find it a bit strong, as most of these patients still showed a negative delta in your figures 3+4, which suggests an improvement, even when it is not huge.

Author Response

Materials & Methods:

Comment a: Line 70-71: a short definition of what is meant by severe obesity would be nice. Is it a specific cut-off at e.g. BMI > 40 or does the CMR have a weight limit?

Answer a: We corrected that sentence and added information about the table weight limit with 250 kg.

Comment b: Line 86 ff: When was the time point of the CMR, as it was mentioned for the echo but not for the CMR itself.

Answer b: We added the time point of CMR to the CMR section. The timing is also described in the Study design section.

Comment c: Line 94: You mention one independent investigator, why not two as you did for echo?

Answer c: We corrected this. PB and MS independently analyzed the CMR data.

Comment d: Line 125 ff: You mentioned using the unpaired t-test for normally distributed values, but did you test for the distribution? And were all data normally distributed? If not, what test was used then?

Answer d: We used for non-normally distributed, unpaired data the Mann-Whitney-U test (i.e., NT proBNP, age). We added this information to the statistical section.

Comment e:      Line 131: It is nice to have a clear description of what the delta values of geometry parameters mean, however, for the LVEF it is the opposite way, which would be nice to be clarified too (mean a negative delta in LVEF indicated a decreasing LVEF).

Answer e: Thank you for this comment. We added the description, how to interpretate delta LVEF values.

Results:

Comment a: Line 149 ff (paragraph 3.1): As you mentioned two independent investigators for the echo data, an interpreter variability should be mentioned.

Answer a: All included echocardiographic variables (LVEDD, LVESD, LVEDV, LVESV, LVEF) are routine echocardiographic parameters obtained during every standard echo study and therefore were associated with high interrater agreement values (intraclass correlation coefficient 0.825–0.937).

Comment b: Line 211 ff (paragraph 3.3): I am not convinced, that this section is a real subgroup analysis for this study. You mentioned the differences in the parameters between AR and FMR, but there is no distinction between these groups regarding the correlation from T1 to the delta measurements. I would like to see an analysis as you did in paragraph 3.2 for AR and FMR separately.

Answer b: Based on the pathophysiological mechanism of volume-overload leading to valvular cardiomyopathy in both study cohorts, we decided to combine both study subgroups in a common pathophysiological entity. Therefore, the correlation analysis included the whole study population.

Separate correlation analysis in both AR and FMR subgroups was not possible in reasonable statistical manner due to the limited sample size.

Discussion:

Comment a: It is quite sad that through the reading process and approval from all authors of the submitted version, no one saw, that lines 233-269 are exactly the same as 273-309

Answer a: We corrected this.

Comment b.  Furthermore, lines 270-272 are information from the template and do not relate to the paper.

Answer b: We deleted the sentences from the template.

Comment c. Line 284: You mentioned that T1 values > 1073ms as “rather fixed defects”. I find it a bit strong, as most of these patients still showed a negative delta in your figures 3+4, which suggests an improvement, even when it is not huge.

Answer c: We rephrased this sentence.

Round 2

Reviewer 1 Report

Manuscript ID: jcm-2061551

Title: Correlation of myocardial native T1 and left ventricular reverse remodeling after valvular surgery

Comments to the authors

1. Line 200(paragraph 3.2): R = 0.29 seems to have a fairly weak correlation. Not remarkably. So, you should delate ‘significantly’.

2. Paragraph 3.2): Surgery yields reverse remodeling of the left ventricle, and T 1 value is useful for predicting this, but in fact, postoperative NTproBNP values should also be described.

3. Paragraph 4): The clinical usefulness of the change in echo parameters from the T1 value should be touched on a little more deeply in the discussion part.

Author Response

Comment 1: Line 200(paragraph 3.2): R = 0.29 seems to have a fairly weak correlation. Not remarkably. So, you should delate ‘significantly’.

Answer 1: We rephrased the sentence. Now, we don't use the word "significantly".

Comment 2: Paragraph 3.2): Surgery yields reverse remodeling of the left ventricle, and T 1 value is useful for predicting this, but in fact, postoperative NTproBNP values should also be described.

Answer 2: That is an interesting aspect. Unfortunately, postoperative NTproBNP values were not assessed postoperatively.

Comment 3: Paragraph 4): The clinical usefulness of the change in echo parameters from the T1 value should be touched on a little more deeply in the discussion part.

Answer 3: We now go into more detail regarding the clinical impact of measurements of T1 values in the discussion section. We added this paragraph: “Repeated measurements of native T1 values in asymptomatic patients with left-sided heart valve insufficiency may support individual decision making in appropriate timing of valvular intervention. Such strategy could prevent irreversible myocardial damage.”

Reviewer 2 Report

Even when most issues were addressed, I still have some comments:

1.       The addition of timing for the CMR study was just added as preoperative, which as you mentioned was already in the study design section mentioned. However, this doesn’t really answer the question. For the preoperative echo you mention that the study was conducted the week before the operation date, but how long before was the CMR study conducted? 1 month, 3 months, 6 months?

2.       For the used statistical tests, it would be nice to know also in the paper which parameter was tested with a ttest or the Mann-Whitney-U.

3.       Thanks for addressing the interrater variability. However, I have two concerns about them. First, how could it be, that the CMR interpretation and the echo have the absolute same values? It is hard to believe. Second, what did you inter-rater variability did you used? The text mentioned the inter-rater agreement (lines 98, 112), but the values and the text in the brackets (lines 112-113) indicate the inter-rater reliability.

4.       I can understand that due to a limited sample size a separate analysis is not possible, but then why the post-hoc subgroup analysis in section 3.3? It is just a description of table 1. Further, I like the colour-coded separation in the figures. However, these let me think, that a separate analysis is much more needed, as I would not be sure, that the T1 cut-off of 1073 ms would fit both groups.

Author Response

Comment 1: The addition of timing for the CMR study was just added as preoperative, which as you mentioned was already in the study design section mentioned. However, this doesn’t really answer the question. For the preoperative echo you mention that the study was conducted the week before the operation date, but how long before was the CMR study conducted? 1 month, 3 months, 6 months?

Answer 1: We added this information. CMR was performed within one week prior to surgery. 

Comment 2. For the used statistical tests, it would be nice to know also in the paper which parameter was tested with a ttest or the Mann-Whitney-U.

Answer 2: We added this information. We marked each variable with an asterix, which was not normally distributed, and which was compared using the Mann-Whitney-U test.

Comment 3: Thanks for addressing the interrater variability. However, I have two concerns about them. First, how could it be, that the CMR interpretation and the echo have the absolute same values? It is hard to believe.

Second, what did you inter-rater variability did you used? The text mentioned the inter-rater agreement (lines 98, 112), but the values and the text in the brackets (lines 112-113) indicate the inter-rater reliability.

Answer 3a: We corrected this. Inter-observer reliability was calculated for CMR.

Answer 3b: By using the values for inter-rater agreement we meant inter-rater reliability. Both terms are often used as synonyms in the statistical literature. We write now “Native T1 measurements showed good inter-observer reliability (intraclass correlation coefficient 0.825–0.937).”

Comment 4: I can understand that due to a limited sample size a separate analysis is not possible, but then why the post-hoc subgroup analysis in section 3.3? It is just a description of table 1. Further, I like the colour-coded separation in the figures. However, these let me think, that a separate analysis is much more needed, as I would not be sure, that the T1 cut-off of 1073 ms would fit both groups.

Answer 4: We share the reviewers concerns about potential differences between FMR and AR patients. However, our sample size analysis revealed a total of 75 patients required to demonstrate a power of 80% with possibility alpha error of 5%. Therefore, both subgroups were too small for separate analysis of primary study endpoint. We aim to perform a separate analysis for FMR and AR patients in an ongoing prospective study of valvular cardiomyopathy in distinct valve pathologies.